# The effect of different extreme weather events on attitudes toward climate change

**Giancarlo Visconti** [ID] [1]☯*, **Kayla Young** [2]☯

**1** Department of Political Science, Pennsylvania State University, University Park, Pennsylvania, United States of America, **2** Department of Political Science, Purdue University, West Lafayette, Indiana, United States of America

☯ These authors contributed equally to this work.
* gvisconti@psu.edu

## Abstract

Can exposure to extreme weather change political opinion and preferences about climate change? There is a growing literature on both the effects of extreme weather events and the factors explaining attitudes toward global warming, though there remains no clear consensus about whether being exposed to extreme weather influences public opinion about climate change. We contribute to this literature by studying the impact of a variety of extreme weather events associated with climate variability, including severe storms, floods, fires, and hurricanes, on attitudes toward climate change. Specifically, we use a three-wave panel survey and a *dynamic* difference-in-differences design to analyze public opinion data at the individual level in the US. We find that exposure to only one extreme weather type—fires—has a small but significant effect on acknowledging the existence of climate change and supporting the need for action. However, that impact quickly vanishes, and other types of extreme weather do not appear to have any effect on opinion.

## Introduction

There is substantial evidence that the global climate is changing and will continue to transform well into the future [1]. Among the most consequential effects of unmitigated climate change is climate variability, which can increase the risks of disasters with potentially devastating impacts [2, 3]. Disasters in the form of extreme weather—including severe storms, floods, fires, and hurricanes—cause major damage across the United States each year. The 2010s were "an unprecedented decade of billion-dollar disasters," with homes and thousands of acres of land engulfed by fires in the West and extreme flooding events inundating towns along the Eastern seaboard [4].

Extreme weather events like these are expected to become even more severe and frequent as climate change progresses in the coming decades. Understanding whether exposure to such disasters changes public opinion and policy preferences is a crucial task, given evidence that mass attitudes can influence policymakers and play a role in determining which policies are ultimately adopted [5, 6]. Exploring factors that influence public opinion about climate change could inform researchers and practitioners focused on policy making in this area and, as a

**Data Availability Statement:** Replication data can be found here: https://dataverse.harvard.edu/dataset.xhtml?persistentId=doi:10.7910/DVN/JIDPDJ.

**Funding:** This project was supported by CONICYT/
FONDECYT 1191522. The funders had no role in
study design, data collection and analysis, decision
to publish, or preparation of the manuscript.

**Competing interests:** The authors have declared
that no competing interests exist.

result, have consequential effects on efforts to curb climate variability. Although the literature on climate attitudes has grown substantially in recent years, several important gaps remain, including a relative lack of causal evidence about the effects of exposure to extreme weather associated with global climate change. Often, these studies also focus on the effect of one type of disaster (e.g., droughts), though many areas are at risk for a range of extreme weather types.

In this study, we aim to offer two key contributions: (i) evidence about the effect of a range of extreme weather types on attitudes about climate change and (ii) an application of a *dynamic* difference-in-difference design (DiD), allowing us to draw inferences of the impacts of extreme weather events. More specifically, we examine the effect of four of the most common disasters associated with climate change in the United States—namely, fires, floods, hurricanes, and severe storms—on individuals' beliefs about the seriousness of global climate change and what should be done to address it. We find that only one type of extreme weather—fires—has a small but statistically significant effect on acknowledging the existence of climate change and supporting the need for action to address the issue. However, this effect disappears in the years after initial exposure. We find no effect for any of the other common extreme weather types. Although these findings do not necessarily settle the debate about how extreme weather events might influence public opinion, they may help to explain some of the mixed results in the literature and offer fruitful areas for future research into particular types of extreme weather.

In the following sections, we offer an overview of existing literature on the impact of extreme weather events, review our data, and describe our research design using a dynamic DiD approach. Next, we report our findings, namely that exposure to fires may briefly impact individuals' opinions about climate change. We conclude by briefly considering the implications of this finding and considering next steps for future research.

## Literature review

Recent surveys demonstrate that the majority of U.S. adults believe global warming is happening and that it is human-caused [7]. There is far less agreement when respondents are asked how concerned they are about potential climate impacts, though, and there is no consensus on the best policy approach for how to address the issue. Extreme weather events associated with a changing climate could serve as a kind of focusing event around climate change, perhaps solidifying public concern that may spur policymakers to act [8]. As extreme weather events become more severe and frequent, it is therefore critical to understand how exposure to a range of disaster types might influence public attitudes and behavior [9, 10].

The literature on the social and political effects of weather conditions, extreme weather events, and climate indicators like temperature changes has grown substantially over the past several decades [11–13]. Many scholars have found that exposure to extreme weather events like tornadoes, hurricanes, floods, and wildfires can influence factors like concern about climate change and support for policy measures to address the issue [14–17]. However, there are also many studies that report null or highly nuanced findings [18, 19]. As stated in a review of this literature, "despite extensive research efforts, the relationship between weather and climate opinions still remains unclear" [20, p.1].

Researchers have used a variety of methodologies to examine the potential impact of extreme weather events. For example, a review of 73 papers prior to 2019 found a substantial diversity in measurement and methodological approaches, which creates challenges for generalizability [20]. Many of these papers draw on survey data [15, 16, 21]; another portion of the literature uses a case-based approach to evaluate the effect of particular disaster events [17, 22, 23]. Studies in this literature also vary by the type and number of extreme weather events they

examine. Some research focuses specifically on extreme droughts [19], evaluates temperature and precipitation anomalies [24], or considers extreme flooding [25]. Other scholars aim to examine a range of weather events that could be associated with climate change [16, 21]. Notably, only some of these papers provide causal evidence [26, 27]. Together, these methodological differences create limitations in the overall understanding of how extreme weather events may influence mass attitudes.

Even though this literature remains unsettled regarding whether and how any given extreme weather event might impact social and political opinions, several important trends have emerged. One is that political ideology can play a critical role in how "evidence" of climate change through an extreme weather event is interpreted. In surveys of individuals who have experienced extreme weather events, political ideology is consistently a major factor in climate policy support, which indicates that motivated reasoning is at play [22, 28]. This is also evidenced in discussions following exposure to extreme weather. In a systematic comparative-case analysis of post-disaster communities, the authors found that residents in Democratic communities were more likely to link a given weather event to climate change [23]. Previously held beliefs and other personal characteristics may be key to understanding how and why certain extreme weather events influence opinions [12, 13]. Other important factors may include the intensity of a given event and the type of policy measure proposed (i.e., mitigation vs. adaptation) [17].

## Research design

Studying the political effects of disasters within a causal framework is complicated by the ever-present potential of hidden biases. Exposure to an extreme weather event could be correlated with a range of political outcomes, which can confound analysis and undermine key inferences [29]. For example, there is evidence that the experience of a major weather event like a hurricane is associated with socioeconomic status [30, 31]. For that reason, many scholars are turning to approaches like a difference-in-differences design to account for potential confounding factors. In this paper, we similarly exploit cross-time and cross-country variation in exposure to extreme weather in the U.S. using a *dynamic* DiD approach.

## Data

We construct an exposure indicator using Federal Emergency Management (FEMA) data for federally declared disasters that may be associated with climate change [32] (see [21] for an example of the use of FEMA data to identify areas exposed to an extreme weather event). This data provides a summary of major disasters and other emergencies beginning in 1953, including severe flooding, volcanic eruptions, landslides, and chemical waste exposures. Data is provided at the county-level and includes the start date of each disaster declaration as well as an incident type (e.g., flood, tornado, etc.). We construct a binary indicator of exposure to an extreme weather event associated with climate change one year before capturing the outcome using panel survey data (years 2010, 2012, and 2014). For example, for the panel wave implemented in 2010, we use FEMA data for 2009. We use this approach since our expectation is that the effect of exposure to a disaster on concerns about climate change will take some time to crystallize since disaster victims usually emphasize concerns about their living conditions just after a disaster rather than paying attention to public or collective concerns [33], such as the impact of climate change. Also, this means that there is a gap between the occurrence of the event and the survey; however, this should not undermine the possibility of finding any effects since previous evidence found that disasters can have long-lasting effects [34]. As a robustness check, we used events that happened the same year as the survey but before its

implementation (see Appendix C in S1 File). The main conclusions of our study hold. Regarding the binary measure, this is required to implement a dynamic DiD design [35].

We identify nine extreme weather types that can be connected to climate change (see Appendix A in S1 File for a discussion). However, not all of these events are equally likely to occur; 88% of the exposed counties were affected by only four types of extreme weather events: severe storms (41%), fires (19%), hurricanes (17%), and floods (11%). The other five extreme weather types are ice storms, mud/landslides, snow storms, tornadoes, and coastal storms. Given their limited prevalence, these do not give us enough data points to implement the DiD with the panel data. As a result, we focus on the four more common types and the ones that provide enough information to estimate exposure effects. Our approach thus diverges from those used in many other studies, which consider only one disaster, such as flooding [36, 37].

We then focus on three sets of subjects based on exposure to extreme weather: never exposed, exposed for the first time (i.e., initial exposure), and exposed one year after initial exposure. Respondents who are considered exposed in the first wave (i.e., always treated) are removed from the analysis since they do not have pre-treatment information [35]. We construct the never-exposed group by subjects not exposed to any extreme weather types that could be associated with climate change. This approach has two main advantages. First, it helps us to have a pure control group and avoid the mistake of including people in the control group who were exposed to a flood when estimating the effect of a fire. Second, it allows us to use the same control group when estimating the effects of exposure to floods, fires, severe storms, and hurricanes.

We use the Cooperative Congressional Election Study (CCES) to construct our dependent variable [38]. This is an online survey conducted by YouGov. The sampling method is matched random sample, which attempts to mimic a representative sample from non-randomly selected pools of respondents. This type of design is ideally suited for online panels [39]. This panel survey data provides an interesting opportunity to explore the effects of extreme weather on public opinion because it includes a question to measure political preferences about climate change that combines acknowledgment of climate change and willingness for action to address this problem. Additionally, its panel structure allows for drawing more credible causal inferences.

There are three waves available for the analysis conducted in the years 2010, 2012, and 2014. Our outcome of interest asks respondents whether they believe global climate change is an established problem and whether action should be taken to address it. The outcome has the following structure: (5) "Global climate change has been established as a serious problem, and immediate action is necessary," (4) "There is enough evidence that climate change is taking place and some action should be taken." (3) "We don't know enough about global climate change, and more research is necessary before we take any actions," (2) "Concern about global climate change is exaggerated. No action is necessary," and (1) "Global climate change is not occurring; this is not a real issue." We use this ordinal variable as our outcome, where values go from 5 to 1 (from both acknowledgment and action to full denial). This data has been used by other scholars as well, in part because it benefits from large samples that are able to represent public opinion across the country and because they use the same wording regarding climate change for each of the measures [16].

## Dynamic difference-in-differences

A DiD design relies on the assumption that unmeasured covariates are either unit-specific but time-invariant or time-specific but unit-invariant. These restrictions imply that the outcomes in each group should (i) differ by the same amount in every period and (ii) exhibit a common set of changes across periods [40]. Therefore, any divergence from these trends can be

interpreted as a treatment effect [41]. Importantly, such a design can strengthen causal claims regarding the relationship between extreme weather and public opinion about climate change and allows us to provide more credible inferences than designs that control for an array of cross-sectional covariates alone.

We use an extension of difference-in-differences designs called dynamic or event study DiD. A common approach to studying disasters' effects would be using a generalized difference-in-differences design (i.e., two-way fixed effects). However, recent research has shown those designs can generate biased estimates when treatment effects are not homogeneous across time [42]. A design that allows us to draw inferences about the impact of disasters but does not present this shortcoming is a dynamic or event study difference-in-differences design [35]. In a dynamic DiD, estimates are aggregated at the length of exposure (e.g., at first exposure), which is particularly helpful when having multiple treatment periods.

We have three waves of panel data in this particular case. Consequently, we can study whether the parallel trend assumption holds in one pre-treatment period and learn the effects of disasters after initial exposure and one year after initial exposure. An event study or dynamic DiD uses a staggered treatment, meaning that when a unit is considered exposed, it stays in that group and does not go back to the control group.

Also, we incorporate key individual-level covariates that should not be affected by exposure to a disaster as controls (i.e., education, gender, and age in the 2010 wave). Covariates were included for efficiency reasons, so they help to reduce standard errors. Findings are consistent when covariates are included (see Appendix B in S1 File). We use bootstrapped-based standard errors, which is recommended when exposure happens in a cluster design [43].

Table 1 provides a summary of the main variables used in the analysis. The outcome follows a 1–5 point scale; the exposure indicators can take four values: 0 means never exposed, 1 exposed for the first time in the first wave, 2 in the second, and 3 in the third. Education, gender, and age are the placebo covariates used in Appendix B in S1 File.

Our design is based on comparing a control and treated group over time. The control group is composed of never-treated respondents (i.e., subjects that were not exposed to a disaster) and the treated group of exposed respondents (i.e., subjects that were exposed to a disaster). Exposure to a disaster can happen at different times, such as during the first, second, or third wave of the panel study. Subjects who live in an exposed county but before exposure are called eventually exposed or eventually treated. For example, a respondent living in a county exposed to a disaster during the 2014 wave would be considered eventually exposed during the 2010 and 2012 waves.

## Results

Fig 1 reports the main results of exposure to the four common types of extreme weather on attitudes toward climate change using the five-point scale version of the outcome for the

**Table 1. Descriptive statistics.**

| Statistic | Mean | St. Dev. | Min | Max |
|---|---|---|---|---|
| Outcome | 3.493 | 1.330 | 1 | 5 |
| First exposure to fire | 0.243 | 0.658 | 0 | 3 |
| First exposure to flood | 0.575 | 1.026 | 0 | 3 |
| First exposure to hurricane | 0.377 | 0.802 | 0 | 3 |
| First exposure to severe storm | 0.647 | 0.911 | 0 | 3 |
| Education | 3.897 | 1.437 | 1 | 6 |
| Gender | 1.445 | 0.497 | 1 | 2 |
| Birth year | 1,954.212 | 11.603 | 1,919 | 1,992 |

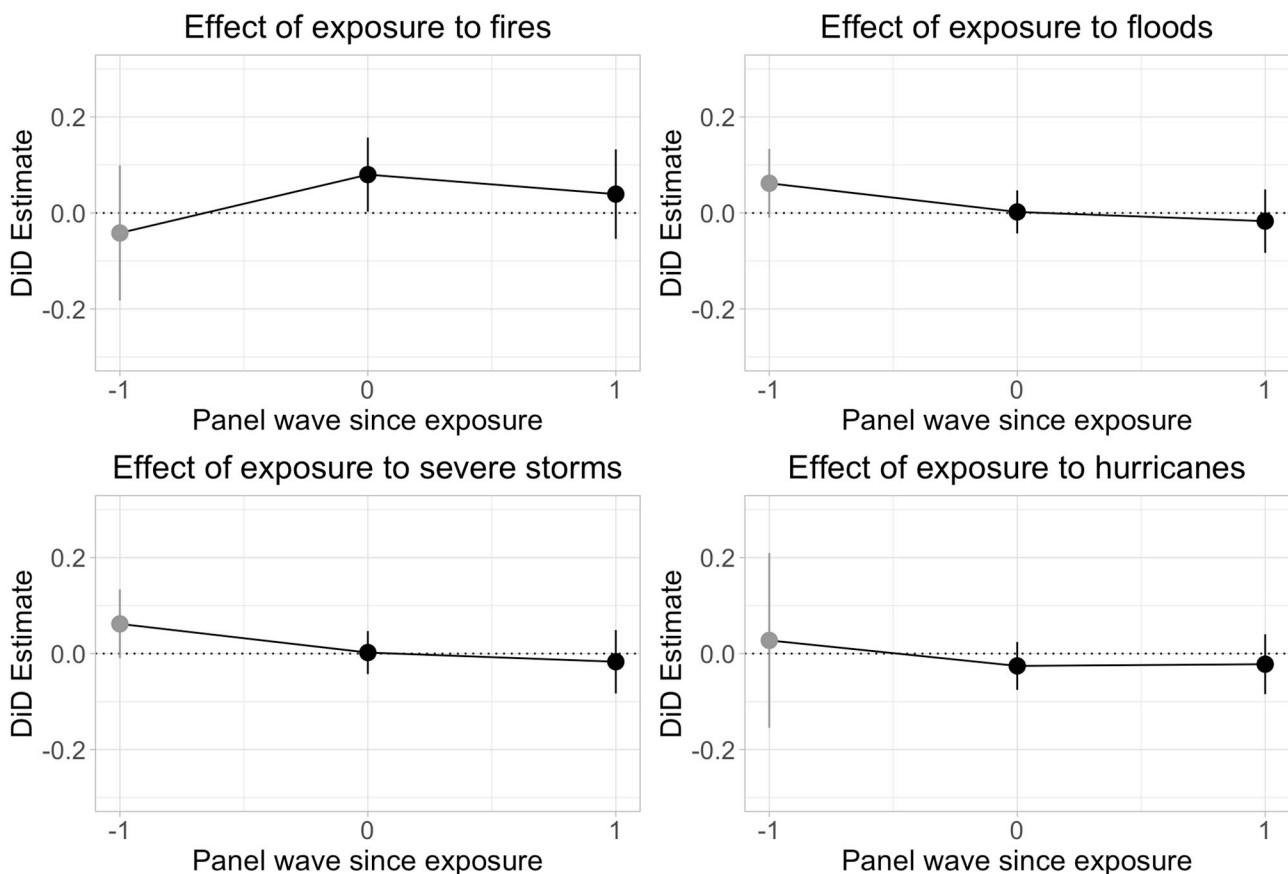

**Fig 1. Effect of exposure to disasters on attitudes toward climate change.** A length of exposure equal to 0 corresponds to the first or initial exposure to a disaster. A length of exposure equal to 1 corresponds to the first period after the initial exposure to a disaster. Conversely, a length of exposure equal to -1 refers to the first period before initial exposure. N = 28,128 (respondents-wave).

different types of disasters. The dots represent the average effects, and the lines the 95% confidence intervals. Results in grey correspond to the pre-exposure analysis, which is the comparison between *never treated* and *eventually treated* (event time = -1). Results in black correspond to the post-exposure analysis, in particular to effects after initial exposure (event time = 0) and one year after initial exposure (event time = 1).

The pre-exposure results for every outcome show no significant effects, which provides crucial evidence for the parallel trend assumption. In other words, the *never treated* and the *eventually treated* group followed a common trend before the latter was exposed to an extreme weather event. Additionally, we observe no significant post-exposure effects at any time for floods, hurricanes, or severe storms on beliefs about the seriousness of global change and policy preferences to address the issue.

As shown in Fig 1a, however, we do see a statistically significant result for first exposure to a fire (event time = 0)—the outcome increases 0.08 points on a 1–5 point-scale (95% CI: [0.002, 0.155]. This effect does not appear enduring, as it disappears one year after initial exposure (event time = 1). We report the results in table format in Appendix D in S1 File. When using a standardized version of the outcome, first exposure to fire (event time = 0) increases the outcome by 0.06 standard deviation units, which is a small-sized effect. Although it is important to recognize these effects are substantively small and short-lasting, they could have

important policy implications. We briefly consider the implications of these findings and suggest steps for future study in the following section.

## Conclusions

As the global climate continues to change, some types of extreme weather events are expected to become more prevalent and severe. Although understanding how these events influence support for climate policy is, therefore, an increasingly important question, the literature remains unsettled about whether extreme weather can influence climate attitudes. We aim to advance this literature by evaluating the effect of an array of common disaster types using a *dynamic* DiD approach. Notably, we find that exposure to a flood, hurricane, or severe storm does not significantly change beliefs about the seriousness of climate change or impact the desire to take policy action to address the issue. However, exposure to a fire does increase support for climate efforts. This effect disappears one year after first exposure, indicating it is not a long-lasting effect.

There are several limitations to our study that are important to consider. For one, our data is limited to a period between 2010–2014. While we recognize that it would be ideal to have more recent data, we believe there are several advantages to examining the impact of extreme weather events on public opinion during this time period. For one, it offers insights prior to the COVID-19 pandemic, which could have shifted priorities and the relative impact of weather events. This data is also prior to the Trump Administration, which significantly rolled back many existing measures to address climate change and may have created a kind of backlash effect. Finally, it's important to recall that extreme weather events are anticipated to become more frequent and severe as climate change progresses. Understanding how extreme weather events may have influenced climate attitudes prior to such environmental and political changes could be particularly useful when compared with more recent data, thereby offering insights into how extreme weather events influence attitudes amidst other pressing developments and when extreme weather is a more common occurrence. Another data limitation is that our measurement of climate opinion is double-barrelled, making it difficult to determine the extent to which the respondents in our sample were expressing higher levels of concern due to fire exposure, stronger beliefs about taking action to address the issue, or both. Future studies should aim to untangle such opinions.

However, we also believe this study has several key implications. First, we demonstrate that not all extreme weather events have the same impact. Therefore, it may be important for scholars to more fully consider the impact of a range of disasters associated with climate change. Our results suggest that fires, in particular, may be an important area for future study. Communities could be exposed to a range of different and more catastrophic weather events in the coming decades. Exploring the causal mechanisms behind such effects continues to be an important area for future study to understand when and how extreme weather events could influence the policy-making process. Questions to consider might include whether the impact of one or more types of extreme weather events are the same in a variety of political contexts, and why the effect of extreme weather like a fire might disappear over time, and what makes an effect more or less long-lasting.

As climate change continues, it may also be helpful to distinguish between support for mitigation (attempting to prevent future effects of climate change) compared to adaptation (adjusting to current and future trends). Increasing wildfire risks have prompted many locales to turn to adaptation measures like power shutoffs to protect public welfare [44, 45]. To avoid such power shutoffs, residents may be more inclined to adopt clean energy options like solar power [46]. Learning from previous studies, it will be important to account for factors like motivated

reasoning by considering political ideology. Recent research has shown that wildfire exposure may affect communities differently based on the political composition and sociodemographic characteristics of communities [47, 48].

Secondly, this study provides an application of a dynamic or event DiD, which allows researchers to apply a DiD approach over several time periods. This could be particularly important for evaluating relationships like the effect of extreme weather types, which we have shown can vary over time. Researchers may want to further expand the use of this approach to understand the relationship between extreme weather and climate attitudes. In addition, this strategy may more accurately illuminate a range of other key relationships like the influence of exposure to a political protest or a conflict event on political behavior.

## Supporting information

**S1 File.**
(PDF)

## Acknowledgments

We thank Thomas Birkland, Rosalee Clawson, Logan Strother, Leigh Raymond, Scott Robinson, Wesley Wehde, and seminar participants at APSA, SPSA, Purdue, Universidad Mayor, and Toronto Political Behaviour Workshop for their valuable comments and suggestions. Authors are listed in alphabetical order. All errors are our own.

## Author Contributions

**Conceptualization:** Giancarlo Visconti, Kayla Young.

**Data curation:** Giancarlo Visconti, Kayla Young.

**Formal analysis:** Giancarlo Visconti, Kayla Young.

**Funding acquisition:** Giancarlo Visconti, Kayla Young.

**Investigation:** Giancarlo Visconti, Kayla Young.

**Methodology:** Giancarlo Visconti, Kayla Young.

**Project administration:** Giancarlo Visconti, Kayla Young.

**Resources:** Giancarlo Visconti, Kayla Young.

**Software:** Giancarlo Visconti, Kayla Young.

**Supervision:** Giancarlo Visconti, Kayla Young.

**Validation:** Giancarlo Visconti, Kayla Young.

**Visualization:** Giancarlo Visconti, Kayla Young.

**Writing – original draft:** Giancarlo Visconti, Kayla Young.

**Writing – review & editing:** Giancarlo Visconti, Kayla Young.

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
