## [Decision Letter · Decision Letter 0]

9 Nov 2023

PONE-D-23-27737The Effect of Different Extreme Weather Events on Attitudes Toward Climate ChangePLOS ONE

Dear Dr. Visconti,

Thank you for submitting your manuscript to PLOS ONE. After careful consideration, we feel that it has merit but does not fully meet PLOS ONE’s publication criteria as it currently stands. Therefore, we invite you to submit a revised version of the manuscript that addresses the points raised during the review process.

Please carefully read and respond to the comments from both reviewers. They provide excellent suggestions on how to address some of the limitations of the manuscript in its current form. 

We look forward to receiving your revised manuscript.

Kind regards,

Kristina Weis Kintziger, PhD, MPH

Academic Editor

PLOS ONE

Journal Requirements:

 "This project was supported by CONICYT/FONDECYT 1191522" 

**Additional Editor Comments:**

The topic of this manuscript has the opportunity to add to the growing body of evidence related to how the public's attitudes on climate change might be impacted due to first-hand experience. I agree with the reviewers' comments on the need for major revision and invite you to respond to these comments and revise your manuscript accordingly.

Reviewers' comments:

Reviewer's Responses to Questions

**Comments to the Author**

1. Is the manuscript technically sound, and do the data support the conclusions?

Reviewer #1: Partly

Reviewer #2: Yes

2. Has the statistical analysis been performed appropriately and rigorously? 

Reviewer #1: N/A

Reviewer #2: I Don't Know

3. Have the authors made all data underlying the findings in their manuscript fully available?

Reviewer #1: Yes

Reviewer #2: No

4. Is the manuscript presented in an intelligible fashion and written in standard English?

Reviewer #1: Yes

Reviewer #2: Yes

5. Review Comments to the Author

Reviewer #1: This paper aims to assess empirically the influence of extreme weather events (EWE) on political opinion and preferences toward climate action. I appreciate the effort to address a highly relevant question in the field. However, there are some important aspects that need further attention to enhance the quality and comprehensiveness of your work. I outline these points below.

1) Literature Review: The paper requires a more comprehensive review of the existing literature concerning the influence of extreme weather events (EWEs) on behavior and attitudes. While you reference studies such as Sloggy et al. (2023), Whitmarch (2008), and Spence et al. (2011), more information is needed to understand their methodologies, findings, and the specific EWEs they investigated. Additional aspects are here also what behavioral dimension they examined (behavior, attitudes, support for specific or general measures, etc.) and the timing they assess the impact of the EWE.

Additionally, several studies that discuss behavioral insights in this domain, such as e.g. Gärtner and Schön (2021) are relevant to your research question but are not mentioned at all.

2) Research Design: Essential details about your research design are missing. such as:

- the nature of the FEMA dataset, who gets surveyed, when, how, and with which survey technique?

- A clarification of the meaning of "eventually treated"?

- Clarification/Justification why data only from 2010, 2012, and 2014 is analyzed. More recent data would have been certainly be better (see point below on validity)

- Elaboration of the statement "estimates are aggregated at the length of exposure". You mean the length of the exposure of a flood that e.g. took place over 3 days?

- How do you account for then for factors like the frequency of exposure and time since exposure – that very likely shape the results?

3) Time Analysis: Since you use a binary indicator of exposure to EWE, a complementary analysis that takes into account the time since exposure is needed. This would allow us to see whether, depending on the specific EWE type, there was an effect at all and how it deteriorated.

4) Discussion on the Relevance of Findings: Include a discussion about the relevance and up-to-dateness of your findings, particularly in the context of political discussions and climate action, and given your data is 10years + old.

Minor Suggestions:

• Update the references in your introduction related to climate variability with more recent sources. Much happened here in the last 10 years!

• Ensure that Figure 1 accurately represents the timing of EWEs and provide clarity on this aspect within the figure description. As far as I understood t=0 is not the EWE but the point of the following survey measurement, which can be many months after.

• Specify the abbreviation "FEMA" when it is first used.

• Include information on the nine disaster types in the main text.

References

Gärtner, L., Schoen, H. Experiencing climate change: revisiting the role of local weather in affecting climate change awareness and related policy preferences. Climatic Change 167, 31 (2021). https://doi.org/10.1007/s10584-021-03176-z

Reviewer #2: Manuscript ID: PONE-D-23-27737

Manuscript Title: The Effect of Different Extreme Weather Events on Attitudes Toward Climate Change

Manuscript Summary:

This study uses panel survey data in the context of a Difference-in-Difference event study analysis to estimate the effects of exposure to a selection of extreme events on climate change beliefs.

In the sections below, I offer several high-level thoughts, followed by specific bulleted questions or comments.

Overall comments:

Thank you for the opportunity to review your study. This paper is well-motivated and the methods are relevant. However, there are considerable opportunities for providing more detailed information about background and methods, as well as connecting the study to the current context.

Motivation/significance:

The paper is well-motivated and fits into the nascent, but growing, literature on the connection between extreme weather events and climate change beliefs, attitudes and policy support. The methods are also appropriate, and to my knowledge, are the first application of diff-in-diff event study (and related inferences) to these questions. In particular, the examination of different event types is an important contribution. Finally, the data are somewhat dated, which affects the degree to which we can apply the findings to current context.

Introduction:

The Introduction offers only a cursory treatment of the relevant literature and relies on outdated citations. It would be useful to add a Background Literature section to expand on these important conceptual frameworks and related findings. For example:

• P. 1 – In the first couple of sentences, more recent citations would be useful – most are over a decade old. The Smith (2020) citation is a good one, although there is a more recent version (Smith, 2023).

• P. 1 – the discussion of the connection between “mass attitudes” and policy change is all of one sentence, even though there has been considerable treatment of these concepts, including the focusing events literature (e.g., Birkland, 2006). Relatedly, the discussion of the theorized relationship between extreme events, public opinion and policy change includes only one reference (Howe et al., 2019) – which is not even the most current review -- despite important seminal articles (e.g., Weber, 2010, 2013), more recent reviews (e.g., Brügger, 2020; Giordono et al., 2020; Howe, 2021) and other empirical contributions to a now-sizable literature (e.g., Boudet et al., 2019; Giordono et al., 2023; Hazlett & Mildenberger, 2020; Zanocco et al., 2019) .

Research Design:

The research design – specifically the analytic approach -- is appropriate and it’s great to see a dynamic difference-in-difference/event study approach applied to panel data to study these questions. However, the narrative needs considerable clarification, and in some places, additional citation. This is especially true for the non-technical reader, who is likely familiar with traditional diff-in-diff, but not necessarily with this approach. Finally, as stated earlier, the data are over a decade old, so the implications of recent changes should be discussed. Specific comments include:

• P. 2 – In the second sentence, there are other relevant citations that could be useful here (more than just a self-cite). See, for example, some of the suggestions above. Seems especially important to describe the strong evidence about the consistent influence of political ideology/party. Also note that these first three sentences, and related citations, would also probably be more appropriate in a thorough Background Literature section. I also wonder whether it would be useful to add hypotheses.

• P. 3 – It might be useful to present both the estimating equation and the parallel trends assumption equation here. On the other hand, it could be alienating to the non-technical reader. Seems like an editorial call.

• P. 3 – Please discuss the inclusion of covariates (e.g., why you included them, even though DiD should, in theory, “difference out” static characteristics). Also indicate which covariates you included.

• P. 3 – Do you use cluster-robust standard errors? Or does the “doubly-robust” estimator (Sant’Anna & Zhao, 2020) negate the need for cluster-robust standard errors?

• P. 4 – it is standard to present your data source first, followed by your analytic approach. And in this case, it is especially important, given that you’re working with panel data. I would strongly recommend switching those sections and editing as necessary.

• P. 4 – I found the Data section (2.2.) a little hard to follow and would like to see more detailed information, as follows:

o I would start with a quick reference to both data sources and/or include a table that shows all variables used in the analysis (including the covariates), along with the source, variable operationalization and descriptive stats.

o Regarding the disaster declaration variables – are these state-level declarations? So someone in a specific state that received federal disaster declaration is assumed to have been "exposed", correct? Or did you use some other type of geospatial matching process to identify exposure? Please clarify, and offer any implications for findings (seems like there’s likely to be considerable measurement error, especially for some event types – what are the likely implications for findings?).

o Relatedly, do we know when in the prior year that the respondent was exposed? Presumably we know when the survey was conducted and when the event occurred. With a binary measure of exposure, you’re lumping everyone in the same group, whether their exposure was 1 month prior or 12 months prior, correct?

o I had trouble understanding the alignment between FEMA declaration data and event exposure. You indicate that you construct your indicator of exposure based on one year before the survey year. So for the panel wave implemented in 2010, you use FEMA data for 2009. And presumably for the 2012 panel, you use FEMA data for 2011. Does that mean you disregard potential exposure in 2010?

o A little more precision around language might help make your process more transparent. For example, “respondents who are exposed in the first wave (i.e., always treated” presumably means respondents who were exposed prior to the first survey wave, rather than “in” it.

o More generally, a visual of the alignment between the FEMA and panel data might be helpful here. In particular, one that shows the relationship between the each declaration year, panel wave and subject group. I could see a flow chart or table performing this function.

• P. 5 – As noted above, I’m concerned about the age of the data. That said, I have some concerns about basing conclusions on data that is approximately a decade old. The authors need to acknowledge this limitation, and describe how older data are relevant to our understanding of current “mass attitudes,” and potential for policy change, given that the American public is currently operating at a higher baseline with respect to beliefs – almost 75% of the American public believes in global warming (Leiserowitz et al., 2023) and experiencing extreme events with a higher frequency and severity (Smith, 2020) than they did a decade ago?

• P. 5 – It’s unfortunate that the CCES questions are double-barreled (asking about beliefs and action in the same question stem and response options). Please discuss the implications for your results.

• P. 5 – On p. 3, you referred to the education, gender and age as the covariates that were incorporated into the analysis. What about political affiliation/orientation? That is a key predictor of climate change beliefs, so it seems important to account for it. If the data simply aren’t available, then that gap should be acknowledged and potential implications explained, or else cite evidence that political affiliation doesn’t typically change over time and therefore will be differenced out in this study.

Results:

While the results appear to be accurate and relevant, the Results section could also use some additional attention:

• P. 5 -- It is standard to provide descriptive statistics for all variables included in the main analysis. See for example, Table 2 in Calloway and Sant’Anna (2020), which is cited in your study. They could either be placed in a broader table in the Data section, as suggested above, or in the Results section. This seems particularly important for understanding the magnitude of the effects.

• P. 5 – In the third sentence, you describe the three groups that correspond to the post-exposure analysis, or time after initial event exposure (-1, 0 and 1). However, I’m still having a hard time understanding what happens to the experience during survey “gap years” (i.e., the years between survey waves – for example, 2011).

• P. 6 – Figure 1a shows the effect estimates, and I like seeing the visual. However, it is standard to also present the regression results in a table, either in the main text or an appendix.

• P. 6 -- It would also be useful to graph the actual data for the various subject groups to give the reader an opportunity to “eyeball” the pre-treatment trend. If you are space-constrained, such a graph could be shown in an appendix. Along those lines, please reiterate to the reader that Figure 1a graphs the effects, not the actual data.

• P. 6 – In figure 1a, the dotted line represents the event time, and the survey was done within 12 months of the event, correct? So the "treatment" occurs between -1 and 0, but we don't know exactly when... this graph makes it look like it occurred just prior to time 0. Doesn't seem like an accurate visualization. Perhaps you should shade that space on the graph to represent the possible range of the event exposure period, or simply take out the dotted line altogether.

• P. 6 -- It might be more precise (and clear) to label the X-axis as “Time to/since event,” rather than “Event time”.

• P. 6 – You indicated earlier that covariates were included. Were any of them significant? (I wouldn’t expect them to be significant, given the research design, but it seems important to add this information). Alternatively, this information could be part of a regression table.

• P. 6 – Please interpret the findings for the non-technical reader. For example, is the magnitude of the statistically significant effects of fire exposure meaningful? Maybe calculate effect size?

• P. 6 – What does the N=28,128 mean? I am presuming that you conducted each event-specific analysis separately, using the same control group but different treatment groups. If that is the case, then shouldn’t the N be different for each treatment group?

Conclusion:

The Conclusion seems thin, with respect to study limitations, implications for political processes and/or policy, and relevant other references. Specifically:

• Please add a paragraph describing the limitations of the study.

• The implications of the main findings for fire exposure should potentially be unpacked a little more, with reference to recent work relating to fire exposure, public opinion and public policy. In addition to the works cited earlier, there are several recent articles that examine support for, and effects of, public safety power shutoffs on beliefs and attitudes that could also be cited (e.g., Guliasi, 2021; Mildenberger et al., 2022; Zanocco et al., 2021, 2022).

References

Birkland, T. A. (2006). Lessons of disaster: Policy change after catastrophic events. Georgetown University Press. http://ebookcentral.proquest.com/lib/osu/detail.action?docID=547783

Boudet, H., Giordono, L., Zanocco, C., Satein, H., & Whitley, H. (2019). Event attribution and partisanship shape local discussion of climate change after extreme weather. Nature Climate Change, 1–8. https://doi.org/10.1038/s41558-019-0641-3

Brügger, A. (2020). Understanding the psychological distance of climate change: The limitations of construal level theory and suggestions for alternative theoretical perspectives. Global Environmental Change, 60, 102023. https://doi.org/10.1016/j.gloenvcha.2019.102023

Giordono, L., Boudet, H., & Gard-Murray, A. (2020). Local adaptation policy responses to extreme weather events. Policy Sciences. https://doi.org/10.1007/s11077-020-09401-3

Giordono, L., Siddiqi, M. U. A., Stelmach, G., Zanocco, C., Flora, J., & Boudet, H. (2023). Trial by Fire: Support for Mitigation and Adaptation Policy after the 2020 Oregon Wildfires.

Guliasi, L. (2021). Toward a political economy of public safety power shutoff: Politics, ideology, and the limits of regulatory choice in California. Energy Research & Social Science, 71, 101842. https://doi.org/10.1016/j.erss.2020.101842

Hazlett, C., & Mildenberger, M. (2020). Wildfire Exposure Increases Pro-Environment Voting within Democratic but Not Republican Areas. American Political Science Review, 1–7. https://doi.org/10.1017/S0003055420000441

Howe, P. D. (2021). Extreme weather experience and climate change opinion. Current Opinion in Behavioral Sciences, 42, 127–131. https://doi.org/10.1016/j.cobeha.2021.05.005

Howe, P. D., Marlon, J. R., Mildenberger, M., & Shield, B. S. (2019). How will climate change shape climate opinion? Environmental Research Letters, 14(11), 113001. https://doi.org/10.1088/1748-9326/ab466a

Leiserowitz, A., Maibach, E. W., Rosenthal, S. A., Kotcher, J. E., Lee, S., Verner, M., Ballew, M. T., Carman, J. P., Myers, T. A., Goldberg, M. H., Badullovich, N., & Marlon, J. R. (2023). Climate Change in the American Mind: Beliefs and Attitudes (Spring 2023). Yale University and George Mason University. https://climatecommunication.yale.edu/wp-content/uploads/2023/09/climate-change-american-mind-beliefs-attitudes-spring-2023.pdf

Mildenberger, M., Howe, P. D., Trachtman, S., Stokes, L. C., & Lubell, M. (2022). The effect of public safety power shut-offs on climate change attitudes and behavioural intentions. Nature Energy, 1–8. https://doi.org/10.1038/s41560-022-01071-0

Sant’Anna, P. H. C., & Zhao, J. (2020). Doubly robust difference-in-differences estimators. Journal of Econometrics, 219(1), 101–122. https://doi.org/10.1016/j.jeconom.2020.06.003

Smith, A. B. (2020). U.S. Billion-dollar Weather and Climate Disasters, 1980—Present (NCEI Accession 0209268) [dataset]. NOAA National Centers for Environmental Information. https://doi.org/10.25921/STKW-7W73

Smith, A. B. (2023). 2022 U.S. billion-dollar weather and climate disasters in historical context | NOAA Climate.gov. National Oceanic and Atmospheric Administration. http://www.climate.gov/news-features/blogs/2022-us-billion-dollar-weather-and-climate-disasters-historical-context

Weber, E. U. (2010). What shapes perceptions of climate change? WIREs Climate Change, 1(3), 332–342. https://doi.org/10.1002/wcc.41

Weber, E. U. (2013). Seeing is believing. Nature Climate Change, 3(4), 312–313. https://doi.org/10.1038/nclimate1859

Zanocco, C., Boudet, H., Nilson, R., & Flora, J. (2019). Personal harm and support for climate change mitigation policies: Evidence from 10 U.S. communities impacted by extreme weather. Global Environmental Change, 59, 101984. https://doi.org/10.1016/j.gloenvcha.2019.101984

Zanocco, C., Flora, J., Rajagopal, R., & Boudet, H. (2021). When the lights go out: Californians’ experience with wildfire-related public safety power shutoffs increases intention to adopt solar and storage. Energy Research & Social Science, 79, 102183. https://doi.org/10.1016/j.erss.2021.102183

Zanocco, C., Stelmach, G., Giordono, L., Flora, J., & Boudet, H. (2022). Poor Air Quality during Wildfires Related to Support for Public Safety Power Shutoffs. Society & Natural Resources, 0(0), 1–15. https://doi.org/10.1080/08941920.2022.2041138

6. PLOS authors have the option to publish the peer review history of their article (what does this mean?). If published, this will include your full peer review and any attached files.

Reviewer #1: No

Reviewer #2: No

---

## [Author Response · Author response to Decision Letter 0]

7 Feb 2024

Please see the memo attached. Thank you.

---

## [Decision Letter · Decision Letter 1]

7 Mar 2024

The Effect of Different Extreme Weather Events on Attitudes Toward Climate Change

PONE-D-23-27737R1

Dear Dr. Visconti,

We’re pleased to inform you that your manuscript has been judged scientifically suitable for publication and will be formally accepted for publication once it meets all outstanding technical requirements.

Kind regards,

Kristina Weis Kintziger, PhD, MPH

Academic Editor

PLOS ONE

Additional Editor Comments (optional):

Thank you for addressing the reviewer comments. We feel that your manuscript is ready for publication.

Reviewers' comments:

Reviewer's Responses to Questions

**Comments to the Author**

1. If the authors have adequately addressed your comments raised in a previous round of review and you feel that this manuscript is now acceptable for publication, you may indicate that here to bypass the “Comments to the Author” section, enter your conflict of interest statement in the “Confidential to Editor” section, and submit your "Accept" recommendation.

Reviewer #2: All comments have been addressed

2. Is the manuscript technically sound, and do the data support the conclusions?

Reviewer #2: Yes

3. Has the statistical analysis been performed appropriately and rigorously? 

Reviewer #2: Yes

4. Have the authors made all data underlying the findings in their manuscript fully available?

Reviewer #2: Yes

5. Is the manuscript presented in an intelligible fashion and written in standard English?

Reviewer #2: Yes

6. Review Comments to the Author

Reviewer #2: The authors have addressed all of my comments satisfactorily, and in my judgment, those of the other reviewer as well. This is an interesting and important article, and I hope that it can be published in short order.

7. PLOS authors have the option to publish the peer review history of their article (what does this mean?). If published, this will include your full peer review and any attached files.

Reviewer #2: No
